# Climate Change Impacts on Soil Erosion and Sediment Delivery to German Federal Waterways: A Case Study of the Elbe Basin

Magdalena Uber [1,*], Ole Rössler [2], Birgit Astor [3], Thomas Hoffmann [1], Kristof Van Oost [4] and Gudrun Hillebrand [1]

1.  Department M3 Fluvial Morphology, Sediment Dynamics and Management, Federal Institute of Hydrology, 56068 Koblenz, Germany
2.  Department M1 Hydrometry and Hydrological Survey, Federal Institute of Hydrology, 56068 Koblenz, Germany
3.  Formerly at Department M3 Fluvial Morphology, Sediment Dynamics and Management, Federal Institute of Hydrology, 56068 Koblenz, Germany
4.  Georges Lemaître Institute for Earth & Climate Research, Earth & Life Institute, Université Catholique de Louvain, 1348 Louvain-la-Neuve, Belgium
*   Correspondence: uber@bafg.de

**Abstract:** Climate change is an important driver of soil erosion and sediment delivery to water bodies. We use observation data from 193 locations in the Elbe River basin as well as spatially distributed erosion rates and sediment delivery simulated in the WaTEM/SEDEM to identify current erosion hotspots and to assess the impact of climate change on future erosion and sediment delivery. We further quantified the uncertainty of the modelling approach by using an ensemble of 21 combinations of global and regional climate models, different emission scenarios and stochastic erosion modelling. Erosion rates are highest on hilly arable land in the central part of the basin as well as in the northeast of Bohemia. Despite considerable differences between climate models and emission scenarios and considerable uncertainties of the erosion model, a future increase in soil erosion and sediment delivery is highly likely. Using the median of climate models and behavioral erosion models, this increase can be up to 14% higher in the far future (2071–2100) than in the reference period (1971–2000) using RCP 8.5. The increase is highest in the Czech part of the basin.

**Keywords:** soil erosion; sediment delivery; modelling; WaTEM/SEDEM; elbe; climate change

## 1. Introduction

Inland waterways serve various purposes. As navigable bodies of water they are an important part of the transport infrastructure, habitat to many species and recreational sites for water sports and leisure. All of these functions are impacted by climate change, for example, via changes in discharge and water temperature. Another aspect is the impact of climate change on soil erosion and delivery of fine-grained sediments into water bodies. These natural processes involve the detachment of fine soil particles, the transport to streams and rivers, transport in the river and finally the deposition in riparian zones or deltas. At a global scale, this flux is estimated to be in the order of $15$–$20 \times 10^9$ t a$^{-1}$ [1]. River systems are highly sensitive to sediment dynamics; thus, excess sediment delivery or depletion can have detrimental effects. While a lack of sediment (caused, e.g., by extraction or upstream damming) can cause channel incision, river bank instability, and saline intrusion in estuaries [2,3], an excess of sediment can cause sedimentation in channels and siltation of reservoirs leading to a loss of reservoir capacity and a need for costly dredging activities [4,5]. The deposition of fine sediments can also cause river bed clogging and a degradation of aquatic ecosystems [6]. Moreover, fine sediments are a preferential transport vector for nutrients and contaminants which can lead to eutrophication or contamination

of water bodies [6–9]. On the other hand, land degradation due to soil erosion threatens biodiversity and natural habitats in uplands [10,11].

Climate is an important driver of soil erosion. The detachment and transport of mineral particles from hillslopes to streams are determined by total rainfall amounts, rain intensity and overland flow. In many river systems, the vast majority of sediment transport occurs during a few heavy precipitation events [12–15]. On a global scale, climate models project an increase in precipitation in a warmer world, but of course, there is a strong spatial and temporal variability [16]. For Germany and the adjacent river basins, total precipitation is projected to increase in winter and decrease in summer with a net increase [17]. Extreme precipitation, which is especially important as a driver of soil erosion, is expected to increase in many parts of the world [16,18–20] and also in Germany and adjacent river basins [17].

Therefore, observations and modelling studies indicate an increase in rainfall erosivity and hence soil erosion [21–27]. Being able to quantify and locate this increase is important for river management and river conservation. Spatially distributed soil erosion models are a valuable tool to assess the impact of climate change on soil erosion and sediment delivery and are increasingly used for this purpose all over the world and at all scales [28]. However, their considerable uncertainty has to be kept in mind and it is crucial to evaluate model output with observational data [29,30]. Several authors have stressed that model evaluation with data from the outlet alone is not sufficient because models can produce good predictions for the wrong reason [30–32]. Using models that consider sediment dynamics within river catchments can help to overcome this problem [33–35]. Several review studies have shown that the predictive capacity of models to reproduce measurements of soil erosion and sediment yield often remains poor [30,36–38]. On the other hand, measurements of soil erosion and sediment delivery are labor intensive, expensive and also subject to large uncertainties (e.g., [12,39]). This leads Alewell et al. [38] to the conclusion that "in bidding farewell to the idea of accurately predicting absolute values with models but rather concentrating on the prediction of relative differences, trends over times and systems reactions to processes and management practices, we can use models as tools to learn about the modelled systems and their reaction". The Water and Tillage Erosion Model and Sediment Delivery Model (WaTEM/SEDEM) [40–42] is a well-established model to simulate the impact of land use change, climate change or soil conservation matters on average annual soil erosion and sediment delivery to water bodies and to identify spatial patterns of soil loss and deposition [28]. The model was chosen for this study because of its suitability for large scale applications and its capability to simulate not only gross soil erosion but also downhill sediment transport and delivery to streams. The Elbe is a major European waterway and crucial for national and international shipping traffic. Historic as well as present-day sediment contamination threaten its water quality and present a challenge for river basin management. The aim of this study is (i) to identify current erosion hotspots in the basin of the Elbe, i.e., a major European river and waterway, based on measured data at 193 measurement sites as well as with distributed modeling with the WaTEM/SEDEM and (ii) to assess the impact of climate change on future soil erosion and sediment delivery to the Elbe. Novel points of this study compared to earlier work in the Elbe basin are a detailed assessment of the uncertainties due to model parameterization and choice of climate models as well as an extensive comparison of modeled results to observed sediment loads at 193 measurement sites.

## 2. Materials and Methods

### 2.1. Study Site

The Elbe is one of the largest rivers in Central Europe (Figure 1). Its basin has a size of 148,300 km$^2$ and is located in Germany (66.5% of the basin) and the Czech Republic (33.7%) with minor parts (less than 1%) in Poland and Austria [43]. It rises at an elevation of about 1400 m above sea level and then traverses Bohemia where the Vltava, which is the main tributary, and the Ohře flow in. At the Czech-German border, it breaks through the Elbe Sandstone Mountains in a narrow valley. About 90 km downstream, the river enters

the North German Plain where the Black Elster, Mulde, Saale and Havel join. The Lower Elbe between the Weir at Geesthacht close to Hamburg and the mouth in the North Sea is influenced by the tides. Land cover in the basin is dominated by agricultural land (45.4%), forests (29.1%) and grassland (11%) [44]).

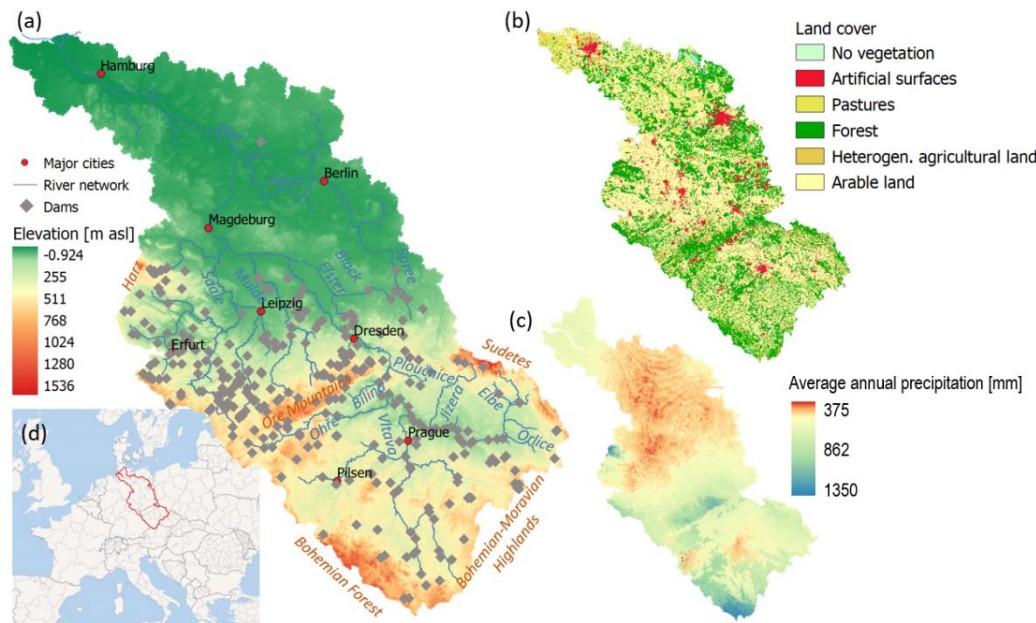

**Figure 1.** The Elbe basin. (**a**) Topography (source: EU-DEM, [45]), (**b**) land cover (reclassified from Corine Land Cover 1990 [46]), (**c**) average annual precipitation (DWD reference ensemble) and (**d**) location of the basin in Central Europe (source: Wikimedia Maps). Figure created by the authors.

The basin is located in the temperate climate zone. Annual average temperatures range from 1–3 °C at the higher altitudes of the low mountain ranges to 8–9 °C in the lowlands [43]. Average annual precipitation ranges from 450 mm in the rain shadow of the mountain ranges to 1700 mm in the Giant Mountains and Jizera Mountains with a spatial mean of 628 mm [43]. The runoff regime is mainly pluvio-nival with a maximum in March and April. In the last decades, floods were also caused by heavy precipitation events in summer such as the ones of August 2002 and June 2013. With an average runoff at Neu Darchau of 861 $m^3 s^{-1}$ (corresponding to 5.4 $l\,s^{-1}\,km^{-2}$) the Elbe is one of the rivers with the lowest runoff rates in Europe [43]. The river and its flow regime are subject to a strong anthropogenic impact via the building of dams, weirs and dikes as well as via changes in the river course. The Elbe is an important European waterway connecting the network of European inland waterways to the port of Hamburg.

The average annual suspended sediment load at Hitzacker is about 600 kt $a^{-1}$. Historically, the Elbe was a highly contaminated river receiving insufficiently treated wastewater from urban centers, industry and agriculture [47,48]. Contaminated legacy sediments from industry and mining are still present in low-flow zones and can be mobilized during floods. These contaminants, as well as present-day sediment contamination from diffuse and point sources, contribute to the fact that the goal of a good ecologic status as defined by the European Water Framework Directive is not achieved [49]. In the Lower Elbe, the estuary and the German Bight, high organic matter loads, eutrophication and oxygen depletion further threaten water quality [50–53].

### 2.2. WaTEM/SEDEM Model

The Water and Tillage Erosion Model and Sediment Delivery Model (WaTEM/SEDEM) [40–42] is a spatially distributed model that simulates average annual soil erosion and sediment de-

livery to water bodies. It is based on the Revised Universal Soil Loss Equation (RUSLE) [54], which calculates average annual soil erosion $E$ [kg m$^{-2}$ a$^{-1}$] as:

$$E = R \times K \times LS \times C \times P \tag{1}$$

where $R$ is the rainfall erosivity factor [MJ mm m$^{-2}$ h$^{-1}$ a$^{-1}$], $K$ is the soil erodibility factor [kg h MJ$^{-1}$ mm$^{-1}$], $LS$ is the slope length factor [-], $C$ is the crop factor [-] and $P$ is the erosion control practice factor [-]. All factors have to be provided as raster maps. $E$ is an estimate of gross or potential erosion. In a second step, the model calculates sediment transport with a transport capacity ($TC$) approach. $TC$ [kg m$^{-1}$ a$^{-1}$] is calculated as:

$$TC = k_{TC} \times R \times K \times \left( LS - 4.08 \, s^{0.8} \right) \tag{2}$$

where $k_{TC}$ is a transport capacity coefficient [m] and $s$ is the slope gradient [m m$^{-1}$]. $k_{TC}$ has to be calibrated. For each cell, soil erosion is added to sediment inputs from upslope cells. It is entirely routed downslope if this sum is lower than the cells transport capacity ($TC$). If this sum exceeds $TC$, downslope transport is limited to $TC$ and the rest gets deposited.

*2.3. Spatial Input Data*

The spatial input data for the WaTEM/SEDEM include elevation data, land use data, soil data and spatially distributed rainfall data. For elevation, the EU-DEM of the European Environmental Agency [45] was used. It has a resolution of 25 m and was aggregated to 100 m for this study to decrease calculation time. The digital elevation model was hydrologically corrected to remove pits and subsequently used to calculate the slope length factor with the equation of McCool et al. [55].

The Corine Land Cover (CLC) data for 1990 [46] was used to calculate the crop factor map. The reclassification of CLC to crop factors was based on the values given by Panagos et al. [56]. On arable land (CLC classes 211 and 212), the crop factor depends on the type of planted crops. As detailed, spatially distributed information on crop types is not available; the crop factor was calculated as the weighted mean of the values proposed by Panagos et al. [56] for the different crop types. Data on crop type composition was obtained from the Federal Statistical Office of Germany [57] and used for the entire basin (including the Czech part). On arable land, the crop factor was further multiplied with a tillage factor $c_{tillage}$ to account for conservation practices. Proposed values for $c_{tillage}$ are 1 for conventional tillage, 0.35 for conservation/ridge tillage and 0.25 for no tillage [56,58,59]. Here we use a value of 0.85 assuming mainly conventional tillage with some areas under conservation tillage.

The soil erodibility factor $K$ was taken from Panagos et al. [60], who provide a soil erodibility map for the EU-25 member states at a resolution of 500 m. It is based on soil survey data at around 20,000 points, the European soil data base and spatial interpolation with location, terrain features and other remotely sensed data [60].

The rainfall erosivity factor $R$ was calculated based on the linear regression of the R-Factor with average annual precipitation [61]. R-Factor maps were calculated for the reference period 1971–2000, the near future (2031–2060) and the far future (2071–2100) with precipitation data obtained from the German Weather Service's (DWD) reference ensemble at a resolution of 0.11° (www.dwd.de/ref-ensemble (accessed on 20 September 2022)). To get a best estimate on future precipitation as well as an estimate of the uncertainty of climate projections, we used the median as well as the 15th and the 85th percentile of the ensemble member projections. The ensemble further provides projections for the three emission scenarios RCP 2.6, RCP 4.5 and RCP 8.5. The list of 21 combinations of global and regional climate models included in the DWD reference ensemble can be found in Table S1 (supplementary material).

A further model input is a map of dams in the basin giving estimates of their sediment trapping efficiency (i.e., the long-term average of the proportion of retained sediment).

Data on the locations of 308 dams in the catchment was taken from the global GOODD data base [62], an inventory of dams in Germany [63], a list of reservoirs in the Czech Republic [64], data provided by the International Commission for the Protection of the River Elbe [44] and OpenStreetMap data (openstreetmap.org, accessed on 10 March 2021 via download.geofabrik.de). Because no data on trapping efficiency is available, it was estimated to be 90% for most dams, which corresponds to the estimates by Junge [65] for the Mulde reservoir. Krasa et al. [64] determined trapping efficiencies for 58 reservoirs in the Czech Republic including many of the largest reservoirs in the Elbe basin. With a few exceptions, the values range between 65% and 99%, the median is 89%. For the weirs in the main channel where water usually flows over the top, this value was reduced to 50%.

*2.4. Suspended Sediment Data*

2.4.1. Measured Data Availability

Suspended sediment concentration (*SSC*) is monitored in the Czech Republic by the Czech Hydrometeorological Institute (ČHMÚ) using daily water samples taken at 14 sites in the Elbe River and its tributaries Orlice, Jizera, Vltava, Ohře, Ploučnice and Bílina. In Germany, water samples are taken at workdays by the Federal Waterways and Shipping Administration (WSV) at 17 measuring sites in the Federal waterways Elbe, Havel, Spree and Saale [66]. Besides these daily measurements, suspended matter is monitored by many of the German Federal States at the measurement sites for water quality assessment with an irregular measurement frequency (weekly to monthly). This data was obtained for >2500 measurement sites in the German part of the Elbe basin. It has to be noted that all measurement networks determine *SSC* gravimetrically and include the mineral as well as the organic material in suspended sediment. *SSC* was estimated indirectly from optical measurements of turbidity at two sites in Bavaria.

A total of 193 measurement sites were selected for the calculation of suspended sediment load and annual yields, based on the availability of discharge data and the length and measurement frequency of the time series. Furthermore, sites that were at a distance of 5 km from another measurement site were excluded, as well as sites where the distribution of discharge at measurement days was not representative for the distribution of discharge in the entire discharge time series. This was tested with a Kolmogorow-Smirnow test. Further information on suspended sediment data and data sources can be found in Table 1.

**Table 1.** Suspended sediment data in the Elbe basin. Sources: (1) International Commission for the Protection of the Elbe River, https://www.ikse-mkol.org/en/themen/die-elbe/zahlentafeln (accessed on 23 February 2021); (2) Czech Hydrometeorological Institute, via personal contact; (3) Federal Institute of Hydrology, Schwebstoffdatenbank SchwebDB; (4) Bayerisches Landesamt für Umwelt, https://www.gkd.bayern.de/de/fluesse/schwebstoff (accessed on 19 April 2021); (5) Senatsverwaltung für Umwelt, Mobilität, Verbraucher-und Klimaschutz, via personal contact; (6) Landesamt für Umwelt Brandenburg, via personal contact; (7) Landesamt für Umwelt, Landwirtschaft und Geologie, https://www.umwelt.sachsen.de/umwelt/infosysteme/ida/ (accessed on 25 March 2021); (8) Landesbetrieb für Hochwasserschutz und Wasserwirtschaft Sachsen-Anhalt, http://www.lhw.sachsen-anhalt.de/gld-portal (accessed on 8 April 2021); (9) Landesamt für Landwirtschaft, Umwelt und ländliche Räume, via personal contact. One measurement site was operated by Brandenburg and Saxony-Anhalt. Further data was obtained from the Elbe Data Information System of the River Basin Community Elbe (https://www.fgg-elbe.de/elbe-datenportal-en.html (accessed on 11 March 2021)); it was combined with the data of the respective federal state.

| Spatial Coverage | Measurement Sites | Selected Sites | Measurement Frequency | Start |
| --- | --- | --- | --- | --- |
| Czech Republic [1] | 10 | 10 | daily | 1993 |
| Czech Republic [2] | 4 | 4 | daily | 2001 |
| Germany [3] | 17 | 14 | on workdays | 1963 |
| Bavaria [4] | 2 | 2 | sub-daily | 2011 |
| Berlin [5] | 6 | 1 | bimonthly-monthly | 1973 |

**Table 1.** *Cont.*

| Spatial Coverage | Measurement Sites | Selected Sites | Measurement Frequency | Start |
|---|---|---|---|---|
| Brandenburg [6] | 444 | 45 | usually monthly | 1989 |
| Saxony [7] | >1000 | 88 | weekly-monthly | 1977 |
| Saxony-Anhalt [8] | >1000 | 26 | Bimonthly-monthly | 2007 |
| Schleswig-Holstein [9] | 4 | 3 | usually monthly | 1991 |

2.4.2. Calculation of Average Annual Suspended Sediment Loads

For stations with daily data of suspended sediment concentration (*SSC*) and discharge (*Q*), daily suspended sediment load ($SSL_d$, i.e., the mass of sediment passing a river cross section per day) can be calculated directly:

$$SSL_d \left[ \text{t d}^{-1} \right] = 0.0864 \times SSC \left[ \text{mg l}^{-1} \right] \times Q \left[ \text{m}^3\text{s}^{-1} \right] \tag{3}$$

Daily loads are summed up over a year to obtain annual suspended sediment load $SSL_a$ [$\text{t a}^{-1}$] and averaged over all available years to obtain average annual loads $\overline{SSL_a}$ [$\text{t a}^{-1}$].

For stations with a weekly or lower measurement frequency, the fact that infrequent measurements may not reproduce the temporal variability of suspended sediment concentrations has to be accounted for. We tested the applicability of sediment rating curves that are used to generate daily time series of *SSC* from daily time series of *Q* based on the *SSC* vs. *Q* relationship in the form of linear, power law and loess regression models. However, the use of sediment rating curves was rejected because of the high scatter in the *SSC* vs. *Q* relationship and the fact that for only 27% of the stations with infrequent measurements any of the models performed substantially better than the base model (i.e., multiplying the time series of *Q* with the mean of *SSC* measurements; criterion: Nash-Sutcliffe efficiency NSE > 0.2). Thus, the discharge-weighted mean concentration method (M18 described by Phillips et al. [67]; ref [12]) was used to calculate annual loads from the daily time series of discharge and infrequent measurements of *SSC*:

$$SSL_a = 0.0864 \frac{\sum_{i=1}^{n} SSC_i Q_i}{\sum_{i=1}^{n} Q_i} \overline{Q_a} \tag{4}$$

where *n* is the number of *SSC* samples in a given year, $SSC_i$ and $Q_i$ are the suspended sediment concentration and discharge at days of sampling and $\overline{Q_a}$ is the mean discharge in the respective year. The standard error of the mean (SEM) was calculated as a measure of uncertainty of the average annual load.

*2.5. Model Calibration and Validation*

WaTEM/SEDEM is mainly calibrated based on the parameter $k_{TC}$ in Equation (2), which governs the calculation of the transport capacity. Other authors have stressed that $k_{TC}$ is highly scale-dependent and has to be calibrated separately for each study site or when input data is changed [41]. Most studies use at least two values for $k_{TC}$; usually a higher value is applied on arable land, vineyards and bare surfaces while a lower value is applied on forests, grassland, shrubland and other natural vegetation classes [41,68–72]. Van Rompaey et al. [73] further suggest to classify $k_{TC}$ based on topography (slope) in catchments with diverse landscapes where they found that a global calibration is not suitable.

Here, we classified $k_{TC}$ based on slope and land use. Thus, four parameters were calibrated, i.e., $k_{TC1}$, $k_{TC2}$ and $k_{TC3}$ for arable land on low, medium and high slopes, respectively, and a factor $f_{kTC,low}$ that is multiplied with $k_{TC1}$, $k_{TC2}$ and $k_{TC3}$ to obtain $k_{TC}$ on non-arable land use for the three slope classes. We generated 1500 random parameter sets of the four parameters with Latin hypercube sampling using the R package lhs [74]. In this way, it was ensured that the entire parameter space is covered equally. $k_{TC1}$, $k_{TC2}$ and $k_{TC3}$ were varied in the range 0–120 m and $f_{kTC,low}$ was varied between 0.05 and 1 to ensure that the entire range of values reported in the literature was included.

The model was run with all 1500 parameter sets, and simulated sediment loads at the 193 measurement sites were compared to loads estimated from measurements. The following objective functions were calculated from simulated and observed values: Nash-Sutcliffe efficiency (NSE), ratio of performance to interquartile distance (RPIQ), mean absolute error (MAE) and the percentage of measurement sites where simulated values of $\overline{SSL_a}$ are within the range of observed $\overline{SSL_a} \pm SEM$ ($P_{uncb}$). While calculating the objective functions, measurement sites with a daily or sub-daily measurement frequency were weighted with a factor ten. This was done to account for the fact that estimates of $\overline{SSL_a}$ obtained from infrequent measurements of SSC are prone to considerable uncertainties while estimates calculated from daily measurements can be considered reliable in catchments of >1000 km$^2$ [12]. Optimal parameter sets were selected based on a multi-objective optimization that seeks to optimize all four objective functions. We used the R package rPref [75], which identifies the set of Pareto optimal solutions, i.e., the parameter sets where no objective function can be made better off without making at least one other objective function worse off. For the subsequent analyses (climate change scenario testing), all Pareto optimal solutions were retained as behavioral parameter sets as proposed in the generalized likelihood uncertainty estimation (GLUE) [76,77]. In this way, a distribution of plausible outputs was obtained instead of deterministic outputs and the uncertainty due to model parameterization could be estimated.

A five-fold cross validation was conducted with the observational data at the 193 measurement sites. The sites were randomly divided into five groups and then iteratively one fifth of the sites was excluded from the calibration, the Pareto optimal parameter sets were identified and the model was evaluated with the remaining fifth of measurement sites. In this way, it could be assessed how the model performs at sites where it was not calibrated.

## 3. Results and Discussion

### 3.1. Observed Avarage Annual Suspended Sediment Loads

The average annual suspended loads in the Elbe catchment are strongly conditioned by the contributing catchment size of the monitoring station and vary between 0.005 and 601 kt a$^{-1}$ ($\overline{SSL_a}$ shown as circles in Figure 2a, supplementary material Figure S2). The average specific load in the entire basin is 4.76 t km$^{-2}$ a$^{-1}$ (Table 2) but specific loads are highly variable (color of subbasins in Figure 2a). Longitudinal changes of $\overline{SSL_a}$ along the Elbe are shown in Figure 2b. The Upper Elbe (13.24 t km$^{-2}$ a$^{-1}$) and the upstream tributaries Orlice (12.77 t km$^{-2}$ a$^{-1}$) and Jizera (12.87 t km$^{-2}$ a$^{-1}$) carry high loads of suspended material (Table 2). At the confluence of the Vltava and the Elbe between the measurement sites Obříství and Dolní Beřkovice, the Elbe and Vltava carry about 110 kt a$^{-1}$ (corresponding to a specific load of 8.05 t km$^{-2}$ a$^{-1}$) and 93 kt a$^{-1}$ (i.e., 3.40 t km$^{-2}$ a$^{-1}$), respectively. The relatively low specific load of the Vltava is due to the large dams of the Vltava cascade that retain large quantities of sediment [64]. Unfortunately, no data is available from the Vltava upstream of the dams or its tributaries. The Ohře river is also dammed and has a relatively low specific load of 3.40 t km$^{-2}$ a$^{-1}$. The specific loads of the tributaries between the mouths of the Ohře and the Mulde, i.e., the Bílina, the Ploučnice, several smaller Saxonian tributaries and the Black Elster range between 5.60 and 7.36 t km$^{-2}$ a$^{-1}$; thus, they are considerably higher than the basin mean at Hitzacker (Table 2). In this part of the Elbe, the high loads at Torgau are remarkable (Figure 2b). As there are no major tributaries in this section, the strong increase between Meissen and Torgau as well as the decrease between Torgau and Wittenberg is not plausible. Thus, we assume that loads at Torgau are overestimated. As the measurement frequency is daily at all measurement sites shown in Figure 2b, we can rule out that this strong overestimation of >100 kt a$^{-1}$ is due to temporal variability of sediment loads. The reasons for this overestimation remain unknown and show the need for a dense network of measurement sites to identify such inconsistencies in the data. The Mulde and Saale rivers contribute 19.93 and 107.66 kt a$^{-1}$, respectively. This corresponds to specific loads below the basin average, but data from upstream measurement sites shows that specific loads can be

much higher there. E.g., at Wurzen it is 24.33 t km$^{-2}$ a$^{-1}$ but much of this sediment is retained in the Mulde reservoir shortly upstream of its confluence with the Elbe river. In the catchment of the Saale, there are also many reservoirs that retain sediment. The northernmost major tributary, the Havel, is a typical lowland river, which has a low specific load of only 0.99 t km$^{-2}$ a$^{-1}$. Furthermore, it can be assumed that a considerable fraction of this load is actually phytoplankton, i.e., algae biomass. Phytoplankton also plays a role in the downstream measurement sites of the Elbe River [78,79]. Hillebrand et al. [80] estimated the share of phytoplankton to total suspended sediment load at Hitzacker to be on average 17.3% with a strong seasonal variability (higher share in summer). Using the same methodology, this fraction is estimated to be approximately 14–40% in the Havel River (unpublished).

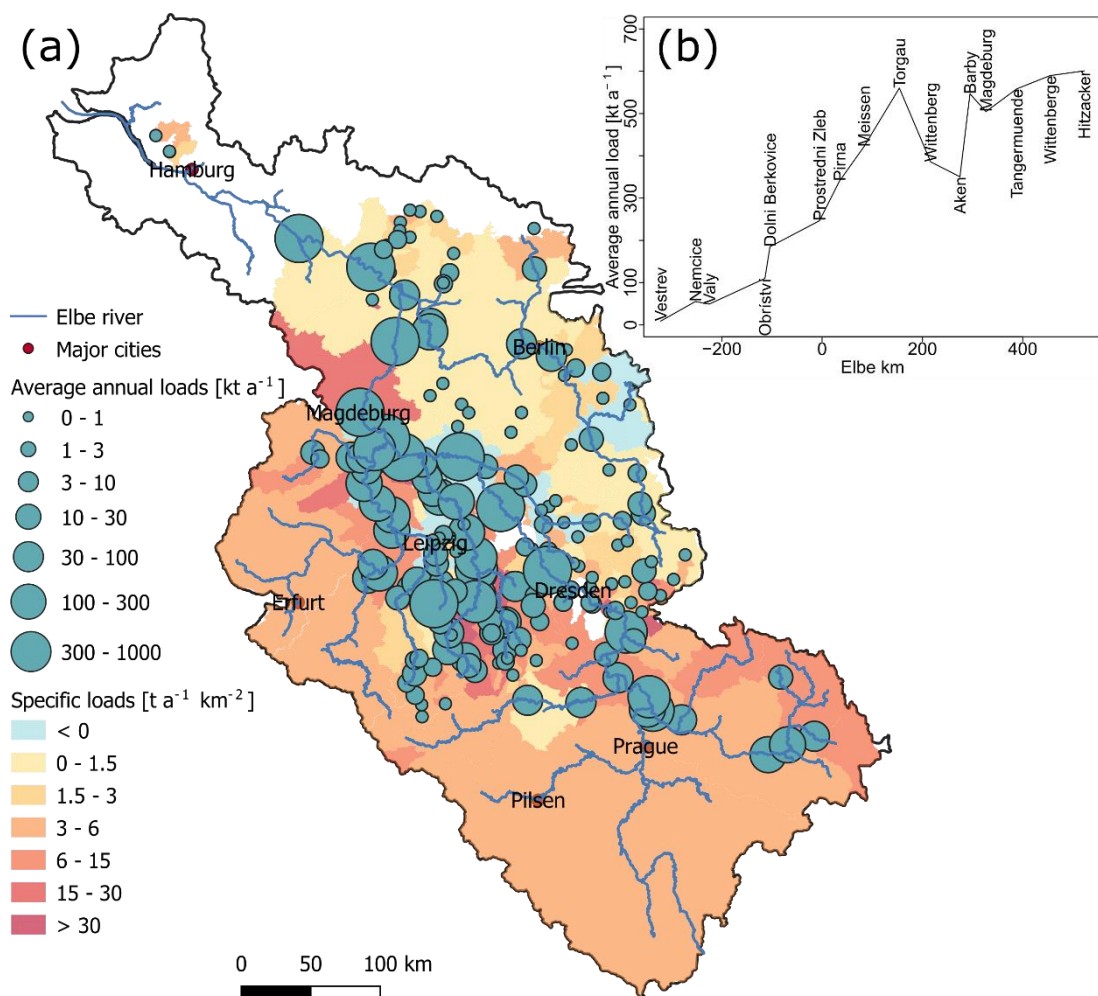

**Figure 2.** (**a**) Spatial distribution of average annual suspended sediment loads (circles) and specific loads (color of subcatchments) in the Elbe basin. (**b**) Longitudinal profile of average annual loads along the Elbe River. The kilometrage corresponds to the system of the German Waterways and Shipping Administration (WSV) where zero corresponds to the Czech-German border.

The high number of measurement sites also allowed us to identify river reaches with negative sediment budgets (i.e., higher inflow of sediment than outflow, Figure 2a). This can indicate deposition, e.g., in reservoirs such as the Mulde reservoir and in floodplains and wetlands such as the Spree Forest, but there are also several reaches where deposition is unlikely and negative sediment budgets rather hint at errors in the data. It can be assumed that the insufficient measurement frequency of many of the measurement sites is a major reason for this error. Moatar et al. [12] estimated that in basins of less than 10,000 km²,

sampling intervals of 3–5 days are required to obtain estimates of $\overline{SSL_a}$ with an error of less than ±20%, while a bi-monthly sampling frequency can lead to errors in the order of ±100%. Consequently, sediment fluxes are underestimated, because the extreme events that carry the majority of sediment loads are likely to be missed [12,39]. This shows the need for a higher measurement frequency to obtain reliable estimates of $\overline{SSL_a}$.

**Table 2.** Average annual suspended sediment loads ($\overline{SSL_a}$) and specific loads of the Elbe and its tributaries.

| River | Location | $\overline{SSL_a}$ $[kt\ a^{-1}]$ | Specific $\overline{SSL_a}$ $[t\ km^{-2} a^{-1}]$ |
|---|---|---|---|
| Upper Elbe | Němčice n. Labem | 55.42 | 13.24 |
| Orlice | Týniště n. Orlicí | 19.05 | 12.77 |
| Jizera | Tuřice | 26.97 | 12.87 |
| Vltava | Zelčín | 93.43 | 3.40 |
| Ohře | Terezín | 16.85 | 3.00 |
| Bílina | Ústí nad Labem | 7.26 | 6.78 |
| Ploučnice | Benešov n. Ploučnicí | 6.49 | 5.60 |
| Saxonian tributaries | - | 16.19 | 7.36 |
| Black Elster | Gorsdorf | 6.07 | 6.07 |
| Mulde | Dessau | 19.93 | 2.78 |
| Saale | Calbe | 107.66 | 4.60 |
| Havel | Rathenow | 34.64 | 0.99 |
| Northern tributaries | - | 4.62 | 1.45 |
| Elbe | Hitzacker | 600.89 | 4.76 |

The comparability of $\overline{SSL_a}$ for various stations may be limited due to the different monitoring periods over which $\overline{SSL_a}$ is averaged. While sediment monitoring at Hitzacker started as early as 1963, data from most other sites were established in the early 1990s or later. Especially, in the presence of trends in *SSC* [81], average annual loads and comparisons between sites depend on the length and period of suspended sediment monitoring.

However, we argue that $\overline{SSL_a}$ vary spatially and in scale over several orders of magnitude and that uncertainties caused by variable monitoring periods and infrequent measurements are of secondary order. Thus, the high density of measurement sites allows an estimate of the spatial distribution of suspended sediment fluxes.

### 3.2. WaTEM/SEDEM Model Calibration and Evaluation

The Pareto optimization based on the objective functions NSE, RPIQ, MAE and $P_{uncb}$ returned 23 optimal parameter sets for the calibrated parameters $k_{TC1}$, $k_{TC2}$, $k_{TC3}$ and $f_{kTC,low}$. Figure 3a shows spatially distributed simulated soil erosion and deposition rates in the Elbe basin in the reference period 1971–2000, and Figure 3c shows averaged rates in the subcatchments. Erosion rates shown here correspond to net erosion, i.e., the difference between on-site erosion and deposition. Figure 3a,c show the 50th percentile, i.e., the median, of the 23 optimal simulations, and Figure 3b,d show quantile maps of the 15th and 85th percentile. The median corresponds to the best estimate, while the quantile maps show the upper and lower boundary of the probable range of values.

The results of the simulations show a clear spatial pattern. In the northern part of Elbe basin, erosion rates are low (subcatchment means of less than 0.009 mm $a^{-1}$ in the Lower Elbe, Middle Elbe, Stepenitz, Havel and Black Elster catchments). In the central parts stretching from the Saale and Mulde catchments via the German part of the Upper Elbe, the Ploučnice and Jizera to the Orlice catchment, erosion rates are highest (subcatchment means of up to 0.029 mm $a^{-1}$). Locally, this rate can exceed 1 cm $a^{-1}$. In the Vltava, Ohře and Czech Upper Elbe catchment, medium erosion rates are found (0.009–0.017 mm $a^{-1}$). This pattern corresponds well to the topography (Figure 1), i.e., erosion rates are highest in hilly terrain. It has to be noted, however, that in altitudes of above approx. 600 m, erosion rates are lower again, because these zones are mainly forested (notably in the Ore Mountains, Harz and Bohemian Forest).

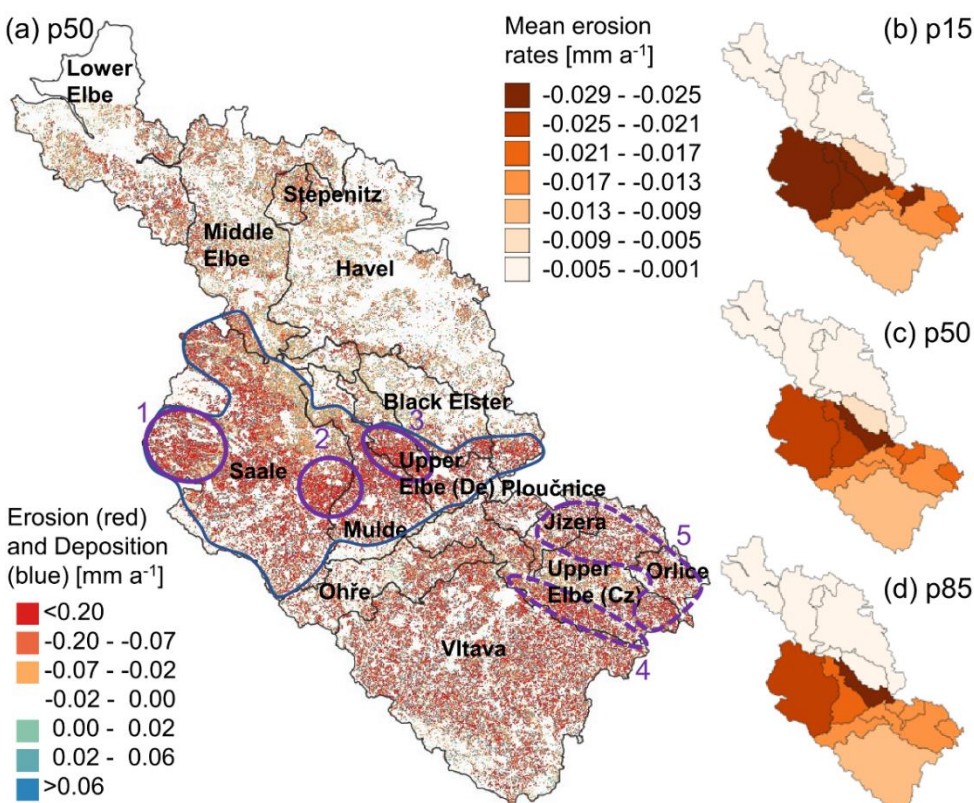

**Figure 3.** (**a**) Spatial distribution of erosion and deposition rates in the Elbe basin modeled with WaTEM/SEDEM. The map shows the 50th percentile of the 23 Pareto optimal simulations. Zones with the highest erosion rates are highlighted in blue and purple; find more information in Section 3.3. (**b**–**d**) Erosion rates averaged for the subcatchments. The maps show the 15th, the 50th and the 85th percentile of all optimal simulations.

A high range of plausible values indicates model imprecision. This is the case, e.g., in the Jizera catchment where the difference between the 85th and the 15th percentile is 60% of the subcatchment median. This percentage is also above 30% for the Stepenitz, the Havel, the Black Elster and the Lower and Middle Elbe subcatchments, while it is low (below 20%) for the Ploučnice, the Orlice, the Vltava, the Ohře, the Upper Elbe and the Saale subcatchments. Thus, with the exception of the Jizera, the imprecision is high in areas of low erosion rates and low in areas of high erosion rates. This is reassuring, as the aim of this modeling study is to identify erosion hotspots and we are less interested in the parts of the catchment where erosion rates are low. Within the optimal parameter sets identified with Pareto optimization, the parameter $k_{TC1}$ is in the range 2–104 m, $k_{TC2}$ is 13–78 m, $k_{TC3}$ is 3–11 m and $f_{kTC,low}$ is 0.35–1. Thus, for the parameter $k_{TC3}$ a clear tendency of good simulations obtained with values in the lower end of the calibration range (1–120 m) can be noted. For the other parameters, optimal parameter sets span a large range within the calibration range, i.e., optimal parameter sets are dispersed within the parameter space and no clear optimum can be identified. The finding that many parameter sets can produce acceptable simulations that reproduce observations equally well is described with the concept of equifinality [82–84] and supports the need of stochastic instead of deterministic modeling [30].

Table 3 shows the minimum, maximum and mean values of the objective functions for simulations run during the calibration and validation steps of the 5-fold cross validation. In general, during validation, the range of the objective function is higher and the mean values are slightly worse than during calibration. The mean objective function values during validation are within the range of values obtained during calibration; thus, the model

does not perform substantially worse during validation and is suitable for application at locations where it was not calibrated.

**Table 3.** Statistics of objective functions obtained during 5-fold cross validation: Nash-Sutcliffe efficiency (NSE), ratio of performance to interquartile distance (RPIQ), mean absolute error (MAE) and the percentage of measurement sites where simulated values of $\overline{SSL_a}$ are within the uncertainty bounds of observed values ($P_{uncb}$).

|  | NSE [-] | RPIQ [-] | MAE [kt a$^{-1}$] | $P_{uncb}$ [%] |
|---|---|---|---|---|
| **Calibration** | | | | |
| Min | 0.51 | 0.78 | 50.329 | 28.788 |
| Max | 0.79 | 2.268 | 85.209 | 42.636 |
| Mean | 0.68 | 1.539 | 63.431 | 35.294 |
| **Validation** | | | | |
| Min | 0.10 | 0.60 | 16.87 | 14.71 |
| Max | 0.96 | 4.07 | 123.40 | 48.39 |
| Mean | 0.59 | 2.10 | 73.07 | 31.93 |

Nonetheless, the values show that the uncertainties are considerable. For example, for none of the simulations, the value of $P_{uncb}$ is above 50%, i.e., at more than half of the measurement sites, the simulated average annual sediment load is outside of the uncertainty bounds of average loads estimated from measurements. Given the inherent uncertainty in soil erosion modeling, the maximum NSE values reported in Table 3 can be considered "good" [85] but absolute errors remain high. The MAE is given in the unit of the simulated variable (here kt a$^{-1}$) and even the minimum during calibration is larger than the mean value of all measurement sites (approx. 41 kt a$^{-1}$). This shows the need to consider more than one objective function.

A direct comparison of observed and simulated values shows that there is a general correlation but also substantial scatter (Figure 4). Furthermore, for some tributaries, there is a systematic error. With a few exceptions, the northern tributaries and the Havel and Spree rivers are systematically underestimated by the model. This can be due to phytoplankton growth, which can be considerable in these lowland rivers and is included in the measurement but not considered by the model. On the other hand, the measurement sites in the Saale catchment, the Saxonian tributaries and some Czech tributaries are overestimated. A possible explanation for the overestimation of these sites with high erosion rates might be that erosion control measures are not sufficiently represented in the model due to a lack of data.

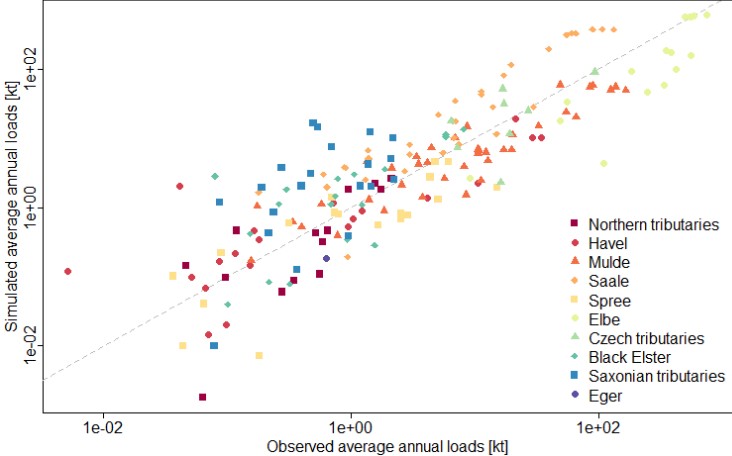

**Figure 4.** Comparison of observed and simulated average annual suspended sediment loads. Each point corresponds to a measurement site in the Elbe basin. The dashed grey line represents the 1:1-line.

In general, the results of the model calibration and evaluation show that the model can reproduce spatial patterns of soil erosion and sediment inputs in the river system but that uncertainties in the absolute loads remain high. Differences between observed and simulated values can well be in the order of 100%. Thus, the model does not succeed in reproducing absolute values but rather gives an estimate of the spatial pattern in the catchment and the order of magnitude of soil erosion and suspended sediment loads in the river system.

### 3.3. Todays Erosion Hotspots

Erosion rates identified with the model are highest in a large zone in the central part of the Elbe basin that stretches over almost the entire Saale catchment, the foothills of the Ore Mountains to the southernmost parts of the Black Elster and Havel catchment in Upper Lusatia (Figure 3a). Within this large zone an erosion hotspot, i.e., a larger, congruent zone of high erosion rates, is located in the Thuringian Basin (marked "1" in Figure 3a). It stretches from the south of the Harz Mountains to the city of Erfurt. On arable land, erosion rates are higher than 0.2 mm $a^{-1}$ in a zone of approx. 3000 km$^2$. The silty loess soils are prone to erosion. A second erosion hotspot is located in the east of the Saale catchment around the Altenburg-Zeitz Loess Hills. A third erosion hotspot comprises the catchments of the left tributaries of the German Upper Elbe between Dresden and Riesa. In the Czech part of the basin, erosion patterns are more scattered over the entire basin and erosion hotspots are less pronounced. Nonetheless, averaged erosion rates are much higher than in the northern part of the basin. Erosion rates are high in the agricultural areas located east of Prague (marked "4" in Figure 3a) and in the foothills of the Sudetes ("5").

Identifying erosion hotspots from the measurement data is challenging because data is available only at distinct points which are not distributed evenly in the basin. Specific loads in units of mass per units of area and time can give valuable information, but they are highly scale dependent. For example, in the erosion hotspot in the Thuringian basin (1), measured data is available only at the outlet of the Unstrut river, where the spatial variability between the erosion hotspot and the low erodible forested mountain ranges is not represented. Here the model can give spatially distributed information where the measured data can only give an integrated signal of the entire subcatchment.

Concerning the erosion hotspot at the Saxonian tributaries (3), the model systematically overestimated measured loads of these rivers (Figure 4). However, it has to be noted, that all these estimates of annual loads are derived from infrequent measurements that lead to a systematic underestimation [12,39]. Thus, it is possible that the model detected an erosion hotspot that could not be identified with the infrequent measurements of suspended sediment concentrations, which are prone to missing the extreme events when a large fraction of annual loads is transported. For the erosion hotspot in the foothills of the Sudetes (5), the model agrees well with the measured data as specific loads estimated from daily *SSC* data are high in the Orlice, Jizera and Upper Elbe catchment at Němčice (Table 2).

The spatial pattern of soil erosion in the Elbe basin was also simulated by Pohlert [86] with the PESERA model. The main erosion hotspot identified in that study stretches from the foothills of the Sudetes to the Orlice catchment. Secondary hotspots were identified in the range of hills in the central part of the basin, stretching from the foothills of the Ore Mountains to Upper Lusatia, in the south of Plzen and in the Saale catchment [86]. Borelli et al. [87] estimated soil erosion with the WaTEM/SEDEM for the European Union. Within the Elbe basin, soil erosion rates are highest in the non-forested Czech part of the basin. High erosion rates are also found in the central part of the basin, in a zone that corresponds well to the zone highlighted in blue in Figure 3a, and even in smaller patches in the northern part of the basin. Thus, the location of erosion hotspots identified here agree fairly well with other studies. However, the magnitude and ranking of erosion rates differ between different studies.

### 3.4. Simulated Future Soil Erosion and Sediment Loads

#### 3.4.1. Changes in Rainfall Erosivity

The median of the climate simulations included in the DWD reference ensemble projects an increase in annual rainfall in the Elbe basin between the reference period (1971–2000) and the near (2031–2060) and far future (2071–2100) for all emission scenarios. This is reflected in an increase in rainfall erosivity of up to 16.7% (Figure 5). However, this increase is not observed in all members of the reference ensemble. In the 15th percentile of all simulations, a decrease in annual rainfall is observed for scenario RCP 2.6 while for the scenarios RCP 4.5 and 8.5 a decrease in the mountain ranges and an increase at the coast in the North German Plain and the Bohemian Basin is observed (Figure S1, Supplementary Material). In the 85th percentile, the increase of rainfall erosivity is locally as high as 26% (Figure S2). Such differences between ensemble members were also reported by other authors [88,89]. While most ensemble members and scenarios are indicative of increased rainfall erosivity, considerable uncertainty in the magnitude of the increase in precipitation in the climate projections remain [17]. To become aware of this uncertainty, it is important to use climate model ensembles, because the results of single models can be biased.

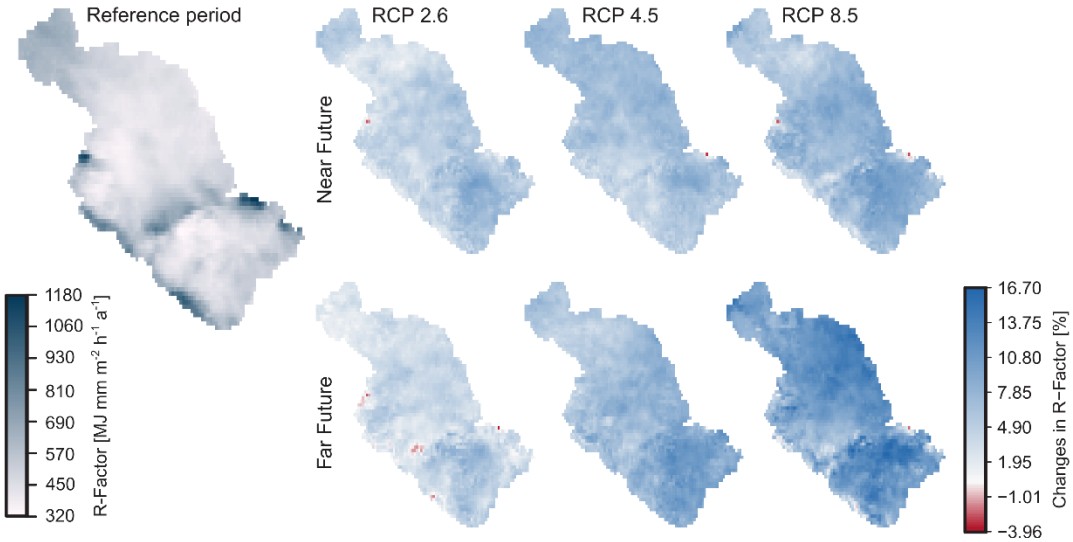

**Figure 5.** Projected changes in rainfall erosivity for the near future (2031–2060) and the far future (2071–2100) with respect to the reference period 1971–2000. The figures show the median of all simulations in the German Weather Service's reference ensemble for the emission scenarios RCP 2.6, 4.5 and 8.5. Maps of the 15th and 85th percentile of the ensemble simulations are shown in Figures S1 and S2 in the supplementary material.

Nonetheless, an increase in rainfall erosivity in the Elbe basin is likely and coherent with the findings of other authors [22,88,90]. For RCP 8.5 and the near future, this increase is most pronounced in the Upper Elbe and the Czech part of the catchment with the exception of the Ohře catchment. In the far future it is most pronounced in the Czech Upper Elbe and its tributaries Orlice, Jizera and Ploučnice, in the Lower Elbe and the Havel catchment.

Here, we calculated rainfall erosivity from average annual precipitation, due to the lack of high-resolution precipitation data for the future. However, rainfall erosivity is closely linked to kinetic energy and thus to rainfall intensity [91,92]. Hence, it varies strongly in space and between different rain events and high spatio-temporal resolution data is needed for a detailed representation of rainfall erosivity [24,93].

We compared the R-Factor used here for the reference period (1971–2000) with (i) the R-Factor map for Germany derived from high spatio-temporal resolution radar data from 2001–2017 by Auerswald et al. [24], (ii) the map for the Czech Republic by Hanel et al. [94] (temporal coverage: 1989–2003) and (iii) the map for the EU by Panagos et al. [95] (covering

1970–2017 with a predominance of the last decade) (Figure 6). The two latter products are derived from high temporal resolution station data and spatial interpolation with covariates of climate, elevation and latitude/longitude. The R-Factor used here correlates well with the R-Factor map by Auerswald et al. [24] ($R^2$ = 0.98), but the values in the latter map are nearly two times higher. This difference might be partially explained by the different time periods, for which both maps are derived, as rain fall erosivity significantly increased between 1970 and 2009 by 60% [21]. Differences between our estimates and those of Hanel et al. [94] and Panagos et al. [95] are much smaller. The slope of the linear regression model between the data used here and the one of Panagos et al. [95] is 1.01 ($R^2$ = 0.95), which is in line with the similar time period used in this study and by Panagos et al. [95] and Hanel et al. [94].

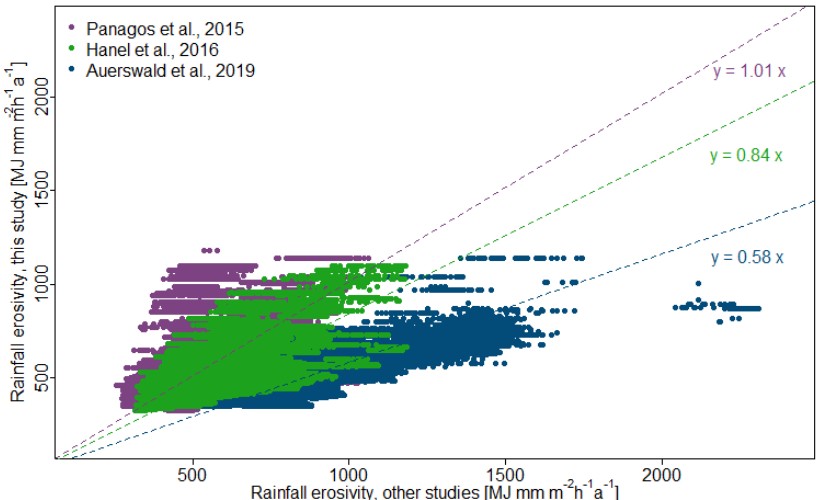

**Figure 6.** Comparison of rainfall erosivity [MJ mm m$^{-2}$ h$^{-1}$ a$^{-1}$] used here, with the data sets generated by other authors (Panagos et al. [95] for Europe, Hanel et al. [94] for the Czech Republic and Auerswald et al. [24] for Germany). Each point corresponds to a cell of the raster maps. The dashed lines show the linear models that were forced to pass through the origin. Figure created by the authors.

In the future, the hydrological cycle is expected to intensify due to warming [96]. This will lead to an increase in frequency and magnitude of heavy precipitation events [18,97–101]. Thus, it is likely that changes in future rainfall erosivity cannot be explained by changes in average annual precipitation alone [102]. The relationship between rainfall erosivity and annual precipitation sum derived from DIN 19708 [61] that we used here is strictly only valid for the period for which the empirical relationship between R and annual precipitation was derived. It is unlikely that this relationship will remain stationary [103]. An increase in frequency and magnitude of intense rainfall events might cause higher rainfall erosivity without a shift of the annual rainfall. Thus, the increases estimated here are conservative and likely underestimate the trend [88].

For future research, we intend to calculate rainfall erosivity from convection-permitting climate simulations. The DWD convection-permitting simulations [104–106] provide precipitation data at hourly temporal and 3 km spatial resolution for the reference period, the near and far future. However, the downside of using convection-permitting simulations is that to date no model ensembles are available.

3.4.2. Changes in Soil Erosion and Sediment Loads

Here, we analyze future changes in soil erosion and sediment loads due to climate change only. We do not consider changes in land use. Increases in soil erosion are relatively low in scenario RCP 2.6 but can be up to 14% in scenario RCP 8.5 for the 50th percentile of climate simulations and the 50th percentile of Pareto optimal erosion models (Figure 7). In

general, changes are lowest in the Lower Elbe and high in the Czech part of the basin. An exception is scenario RCP 8.5 where the catchments of the Havel and Black Elster are also strongly affected.

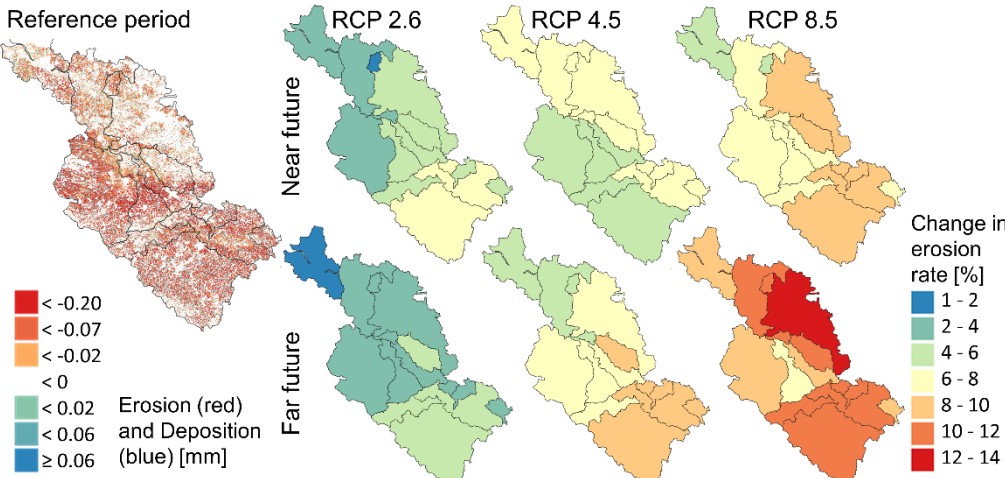

**Figure 7.** Projected changes in erosion rates for the near future (2031–2060) and the far future (2071–2100) with respect to the reference period (1971–2000). The map shows the 50th percentile of the 23 Pareto optimal simulations run with rainfall erosivity derived from the 50th percentile of average annual precipitation obtained from the DWD reference ensemble of climate simulations.

The uncertainty of emission scenarios and climate projections is propagated and amplified with the uncertainty of soil erosion modeling. Thus, the ranges of values obtained with the entire range of 23 Pareto optimal simulations run each with the 15th, 50th and 85th percentile of future rainfall erosivity and the emission scenarios RCP 2.6, RCP 4.5 and RCP 8.5 are very large (Figure 8). For example, at the Elbe at Hitzacker, simulated loads in the far future range between 558 and 759 kt a$^{-1}$. For all sites, some simulations show a decrease while others show an increase with respect to the median of simulated values for the reference period (Figure 8). Nonetheless, the median is increasing at all sites, indicating that an increase in future sediment yields is likely. Ambiguous results in the trend of future sediment loads due to different projections by different climate models were also obtained by Pohlert [86], who simulated erosion and sediment yields in the Elbe catchment with the PESERA model and climate data from five coupled runs of global and regional climate models.

Even though many studies found an increase in precipitation extremes and rainfall erosivity in the last decades in Central Europe [20,23,24,98,107–109], observed suspended sediment concentrations and loads throughout Germany show a decreasing trend between 1990 and 2010 [81]. This unexpected observation cannot be explained by increasing retention because the construction of large reservoirs and dams ceased in the 1980s, while the most pronounced decrease in sediment concentrations occurred between 1995 and 2005 [81]. Hence, Hoffmann et al. [81] assume that the introduction of conservation agriculture and reduced sediment connectivity due to the construction of local features such as rainwater retention basins are the most likely reason for the observed trend. Because such local measures are difficult to quantify at larger scales, there is a need for further research to investigate the reasons for this decreasing trend despite an increase in rainfall erosivity. Between 2010 and 2020, sediment concentrations remained more or less stable, and it is entirely possible that the observed decreasing trend will shift towards an increasing trend in suspended sediment concentrations in an intensified hydrological regime. While an increase in suspended sediment concentrations does not hinder navigability of the waterway, it can lead to increased costs for dredging in zones of low flow velocity, a decrease in water quality and impairments for renaturation measures due to siltation. Thus, continued efforts in erosion control and conservation farming are very important to reduce the impact

of climate change on soil erosion and associated deliveries of suspended sediment as well as particle bound nutrients and contaminants to water bodies. While our study indicated a first approximation of likely changes due to rainfall erosivity, future modelling exercises including extreme precipitation events, land use scenarios and scenarios for the management of stormwater are urgently required.

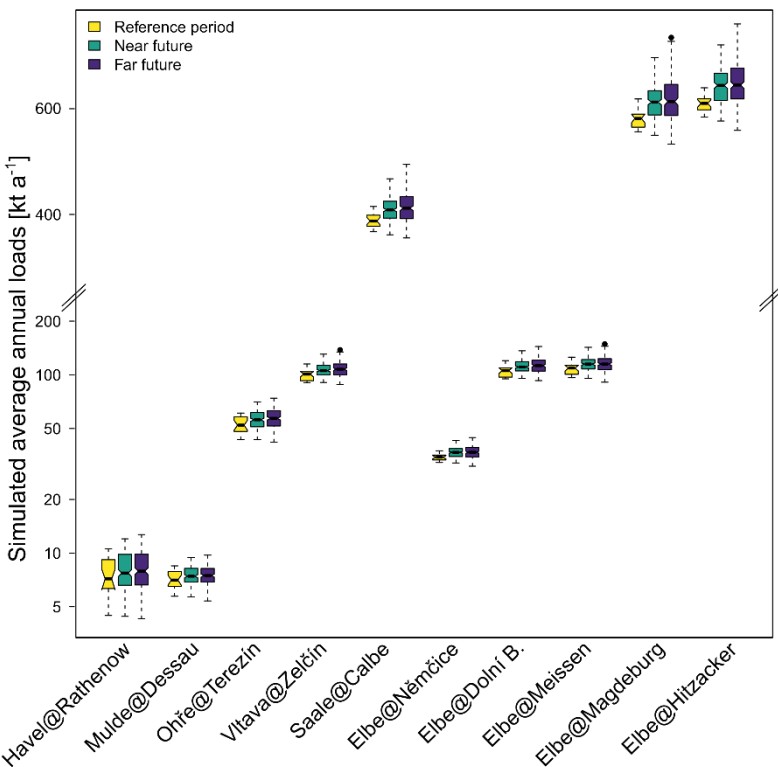

**Figure 8.** Boxplot of simulated average annual loads at different measurement sites along the Elbe and its tributaries. For the reference period, the range of values obtained with the 23 Pareto optimal erosion models is given, for the near and far future; all realizations of the 23 Pareto optimal solutions run with the 15th, 50th, and 85th percentile of climate models for RCP 2.6, RCP 4.5 and RCP 8.5 are represented. Note that for a better visualization, the y-axis is log scaled up to 200 kt $\text{a}^{-1}$ only.

## 4. Conclusions

We used measured data as well as distributed numerical modeling to identify erosion hotspots in the Elbe basin and to assess the impact of climate change on future soil erosion and sediment delivery to water bodies. From this work, we can draw several conclusions:

- Even though they are prone to substantial errors, infrequent measurement of suspended sediment concentrations at numerous water quality assessment sites can give an estimate of spatial patterns of soil erosion and sediment delivery.
- Distributed modeling of soil erosion and sediment delivery with the WaTEM/SEDEM is very helpful to identify spatial patterns of erosion rates within large basins. Nonetheless, it is subject to considerable uncertainties.
- Uncertainties in simulated erosion rates and sediment loads associated to model parameterization are inevitable. For simulated mean erosion rates of single subbasins it was up to 60%. This uncertainty can be assessed with stochastic modeling.
- Further uncertainties about future changes in rainfall erosivity are due to differences between single members of the climate model ensemble used here and between the emission scenarios. To assess this uncertainty, it is important to use climate model ensembles instead of the output of a single model.

- Major erosion hotspots are located in the central part of the basin, in a zone stretching from the Saale catchment via the foothills of the Ore Mountains to Upper Lusatia, as well as in the foothills of the Sudetes in the northeast of the Czech part of the basin.
- Despite the uncertainties in erosion modeling, it is very likely that future erosion and sediment delivery will increase (mainly in the southeastern part of the basin) but the absolute values are highly uncertain and depend strongly on future emissions.
- Further research is needed to assess the role of erosion control practices and sediment retention measures as well as the impact of the likely future increase in extreme precipitation on future soil erosion rates.

**Supplementary Materials:** The following supporting information can be downloaded at: https://www.mdpi.com/article/10.3390/atmos13111752/s1, Figure S1: Changes in rainfall erosivity for the 15th percentile of climate models; Figure S2: Changes in rainfall erosivity for the 85th percentile of climate models; Table S1: List of ensemble members included in the German Weather Service (DWD) reference ensemble. Average_annual_loads.xlsx: File providing metadata of SSC measurement sites and average annual loads.

**Author Contributions:** Conceptualization, T.H., K.V.O. and G.H.; formal analysis, M.U., O.R. and B.A.; funding acquisition, G.H.; methodology, M.U., O.R., B.A. and G.H.; project administration, T.H. and G.H.; software, M.U., O.R. and K.V.O.; supervision, G.H.; visualization, M.U.; writing—-original draft, M.U.; writing—-review and editing, T.H. and G.H. All authors have read and agreed to the published version of the manuscript.

**Funding:** This research was funded by the German Federal Ministry for Digital and Transport Network of Experts.

**Data Availability Statement:** Data on average annual suspended sediment loads estimated for 193 measurement sites is provided in the supplementary material. The EU-DEM and Corine Land Cover data are publicly available at https://land.copernicus.eu/imagery-in-situ/eu-dem/eu-dem-v1.1 (accessed on 21 September 2022) and https://land.copernicus.eu/pan-european/corine-land-cover/clc-1990 (accessed on 21 September 2022). The K-factor map provided by Panagos et al. [60] can be requested at https://esdac.jrc.ec.europa.eu/content/soil-erodibility-k-factor-high-resolution-dataset-europe (accessed on 21 September 2022). Hydrometeorological data from the DWD reference ensemble can be obtained via the DAS Basisdienst (www.das-basisdienst.de (accessed on 21 September 2022)).

**Acknowledgments:** We want to thank our colleagues at Federal Institute of Hydrology and the members of the Network of Experts Themenfeld 1 for the fruitful discussions. The suspended sediment data used here was provided by (i) the suspended sediment monitoring network of the German waterways established and maintained by the Federal Waterways and Shipping Administration (WSV), (ii) the Czech Hydrometeorological Institute (ČHMÚ), (iii) the environmental agencies of the German federal states mentioned in Table 1 and (iv) the International Commission for the Protection of the Elbe River. We want to thank all involved staff for the provision of the data and help with its acquisition. We further thank the editor and two reviewers for their constructive feedback that helped to improve the quality of this article.

**Conflicts of Interest:** The authors declare no conflict of interest.

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
