# Peer review of "Climate Change Impacts on Soil Erosion and Sediment Delivery to German Federal Waterways: A Case Study of the Elbe Basin"

_atmosphere, doi:10.3390/atmos13111752_

Round 1

Reviewer 1 Report (Previous Reviewer 1)

The article is now accepted

Reviewer 2 Report (Previous Reviewer 2)

The manuscript is very well written and the topic is quite pertinent as it may have transboundary effects.

This manuscript is a resubmission of an earlier submission. The following is a list of the peer review reports and author responses from that submission.

Round 1

Reviewer 1 Report

The article deals with the climate change impacts on soil erosion and sediment delivery to German Federal waterways. In General, it was well structure and interesting for the journal’s readers. However, some improvements are necessary so as to be accepted.

In the abstract some information about future climate conditions are missing. Do you use RCMs or GCMs? Which version and what resolution? CORDEX? It must be clarified.

Line 45. Nevertheless, the water erosion has been emerged as a key factor of land degradation that threatens biodiversity and natural habitats survival (Stefanidis et al., 2022; Orgiazzi and Panagos 2018)

Stefanidis, S., Alexandridis, V., & Ghosal, K. (2022). Assessment of Water-Induced Soil Erosion as a Threat to Natura 2000 Protected Areas in Crete Island, Greece. Sustainability, 14(5), 2738.

Orgiazzi, A., & Panagos, P. (2018). Soil biodiversity and soil erosion: It is time to get married: Adding an earthworm factor to soil erosion modelling. Global Ecology and Biogeography, 27(10), 1155-1167.

Line 55. To that end increases in the mean annual soil loss rate has been reported in European country at watershed-scale studies (Stefanidis and Stathis 2018; Panagos et al. 2020)

Stefanidis, S., & Stathis, D. (2018). Effect of climate change on soil erosion in a mountainous Mediterranean catchment (Central Pindus, Greece). Water, 10(10), 1469.

Panagos, P., Ballabio, C., Himics, M., Scarpa, S., Matthews, F., Bogonos, M., ... & Borrelli, P. (2021). Projections of soil loss by water erosion in Europe by 2050. Environmental Science & Policy, 124, 380-392.

Line 74-75. What are the advantages of using WaTEM/SEDEM rather than other well documented erosion prediction model (RUSLE, Gavrilovic ect.) The authors must justify the selection of the model.

In the last paragraph of the introduction clearly state the novel points of the current research

Better and more in depth explain the statistical analysis and model calibration. Also discuss/compare the findings with other similar researches.

Some thoughts for future research that derived from the current analysis it would be useful for the readers. Please add

This is an interesting article and highly important for the readers of the journal. My suggestion in the aforementioned parts I believe will strengthen the sound of the research and better explain to the reader the importance of such analysis.

Reviewer 2 Report

Page 10, line 395: minimum, maximum

Page 12, line 451: in the agricultural areas located at east of

Page 13, line 512: data for the - the sentences is not finished

Conclusions  - line 611: patterns

Conclusions should be improved. One main issue is uncertainty. Authors should elaborate more on this and how to overcome it.

References should be more updated (there are 35 references with more than 10 years).